# The Safe Values of Quantitative Perfusion Parameters of ICG Angiography Based on Tissue Oxygenation of Hyperspectral Imaging for Laparoscopic Colorectal Surgery: A Prospective Observational Study

**DOI:** 10.3390/biomedicines11072029

**Published:** 2023-07-19

**Authors:** Gyung Mo Son, Armaan M. Nazir, Mi Sook Yun, In Young Lee, Sun Bin Im, Jae Yeong Kwak, Sang-Ho Park, Kwang-Ryul Baek, Ines Gockel

**Affiliations:** 1Department of Surgery, Pusan National University Yangsan Hospital, Pusan National University School of Medicine, Yangsan 50612, Republic of Korea; 2Research Institute for Convergence of Biomedical Science and Technology, Pusan National University Yangsan Hospital, Yangsan 50612, Republic of Korea; msyun@pusan.ac.kr (M.S.Y.); vmffkdl38@naver.com (I.Y.L.); 3School of Medicine, Royal College of Surgeons in Ireland, University of Medicine and Health Sciences, D02 YN77 Dublin, Ireland; armaannazir15@gmail.com; 4Department of Medicine, Pusan National University School of Medicine, Yangsan 50612, Republic of Korea; brandonim@naver.com; 5Department of Electronics Engineering, Pusan National University, Busan 46241, Republic of Korea; gjy2022@pusan.ac.kr (J.Y.K.); propia@pusan.ac.kr (S.-H.P.); krbaek@pusan.ac.kr (K.-R.B.); 6Department of Visceral, Transplantation, Thoracic and Vascular Surgery, University Hospital Leipzig, 04103 Leipzig, Germany; ines.gockel@medizin.uni-leipzig.de

**Keywords:** perfusion imaging, oxygen saturation, indocyanine green, colorectal surgery, surgical anastomosis, laparoscopy

## Abstract

Background: Safe values for quantitative perfusion parameters of indocyanine green (ICG) angiography have not been fully defined, and interpretation remains at the surgeon’s discretion. This prospective observational study aimed to establish the safe values for the quantitative perfusion parameters by comparing tissue oxygenation levels from HSI images in laparoscopic colorectal surgery. Methods: ICG angiography was performed using a laparoscopic near-infrared (NIR) camera system with ICG diluted in 10 mL of distilled water. For quantitative perfusion parameters, the changes in fluorescence intensity with perfusion times were analyzed to plot a time–fluorescence intensity graph. To assess real-time tissue oxygen saturation (StO_2_) in the colon, the TIVITA^®^ Tissue System was utilized for hyperspectral imaging (HSI) acquisition. The StO_2_ levels were compared with the quantitative perfusion parameters derived from ICG angiography at corresponding points to define the safe range of ICG parameters reflecting good tissue oxygenation. Results: In the regression analysis, T_1/2MAX_, T_MAX_, slope, and NIR perfusion index were correlated with tissue oxygen saturation. Using this regression model, the cutoff values of quantitative perfusion parameters were calculated as T_1/2MAX_ ≤ 10 s, T_MAX_ ≤ 30 s, slope ≥ 5, and NIR perfusion index ≥50, which best reflected colon StO_2_ higher than 60%. Diagnostic values were analyzed to predict colon StO_2_ of 60% or more, and the ICG perfusion parameters T_1/2MAX_, T_MAX_, and perfusion TR showed high sensitivity values of 97% or more, indicating their ability to correctly identify cases with acceptable StO_2_. Conclusion: The safe values for quantitative perfusion parameters derived from ICG angiography were T_1/2MAX_ ≤ 10 s and T_MAX_ ≤ 30 s, which were associated with colon tissue oxygenation levels higher than 60% in the laparoscopic colorectal surgery.

## 1. Introduction

Indocyanine green (ICG) angiography can be used to assess colonic perfusion status and predict whether sufficient blood flow can be delivered to the tissue through the collateral circuit after vascular ligation during colorectal surgery [1]. If the blood flow in the colon segment is poor, the risk of anastomotic complications could increase after surgery [2]. Using intraoperative perfusion assessments, surgeons can identify poorly perfused or ischemic regions and decide to adjust the transection line to avoid poorly perfused areas at the anastomosis site to optimize surgical outcomes in colorectal surgery [3].

The introduction of ICG fluorescence imaging has provided an objective assessment of bowel perfusion in the surgical field [4]. However, quantitative analysis of ICG perfusion parameters still requires clinical validation and standardization. Safe values for quantitative perfusion parameters reflecting good perfusion have not been fully defined, and interpretation of the data remains at the surgeon’s discretion. Furthermore, the range of quantitative perfusion parameters obtained from ICG angiography varies depending on the fluorescence camera system, angiography protocol, external conditions, and the patient’s characteristics [5].

Hyperspectral imaging (HSI) is a technique that can be used to analyze wavelengths of light and provide detailed information regarding tissue oxygenation levels. By analyzing the spectral response of tissues across a range of wavelengths, non-invasive and real-time assessment of tissue oxygenation can be achieved with a short wait time of around 15 s. This technology has potential applications in a range of medical fields, including colorectal surgery [6].

By analyzing the ICG perfusion parameters in conjunction with tissue oxygenation levels obtained from HSI image, “safe values” for the ICG perfusion parameters could be established [7,8,9]. These safe values will serve as standardized guidelines, enabling clinicians to interpret the quantitative perfusion parameters of ICG angiography more effectively and consistently when assessing the perfusion status of the colon. This standardized perfusion assessment would enhance the clinical utility of ICG angiography for assessing colon perfusion, potentially leading to more informed surgical decision making and improved clinical outcomes [10].

This prospective observational study aimed to establish the safe values for the quantitative perfusion parameters of ICG angiography by comparing tissue oxygenation levels from HSI images in the laparoscopic colorectal surgery.

## 2. Materials and Methods

### 2.1. Patients

This prospective observational study included 68 patients with colorectal cancer between January 2021 and December 2022. The inclusion criteria were patients aged 19–80 years who had sigmoid colon or rectal cancer and underwent laparoscopic surgery with or without diverting ileostomy. The exclusion criteria were hemodynamic instability, emergency surgery, and pregnancy. All patients enrolled in the study were screened for any history of allergies or adverse effects to either the contrast agent used for computed tomography (CT) or iodine-containing drugs used to reduce the risk of cross-reaction with ICG. Written informed consent was obtained from all patients included in this study. The study was conducted after receiving approval from the Institutional Review Board of the Pusan National University Yangsan Hospital.

### 2.2. ICG Angiography

ICG angiography was performed using a laparoscopic near-infrared (NIR) camera system (1588 AIM camera system, Stryker, Kalamazoo, MI, USA) with ICG (25 mg, Daiichi Sankyo, Tokyo, Japan) diluted in 10 mL of distilled water. The minimum dose was 0.2 mg/kg, which was injected into the peripheral vein of the forearm. After the injection, 20 mL of normal saline was infused to flush the injection line. Colonic perfusion was monitored for 2 min using the endoscopic NIR visualization (ENV) mode, and a fluorescence perfusion video was obtained. For quantitative perfusion parameters, the changes in fluorescence intensity with perfusion times were analyzed to plot a time–fluorescence intensity graph using ICG Analyzer Program 8.0 (designed by the Microprocessor Application Laboratory, Pusan National University, Busan, Republic of Korea) [11]. The perfusion time factors were calculated as T_MAX_ (time from first fluorescence increase to maximum), T_1/2MAX_ (time to reach half of the maximum), and perfusion TR (time ratio, T_1/2MAX_/T_MAX_). The fluorescence intensity factors were F_MAX_ (maximal intensity) and slope (calculated as ΔF/ΔT = F_MAX_/T_MAX_) (Figure 1). During ICG angiography, a standardized protocol was applied to minimize the impact of external factors and improve the accuracy and reliability of the measurements of fluorescence intensity. In this study, the surrounding room lights were turned off, and ICG-specific ENV modes with a laser source were used. Keeping a constant distance of 4–5 cm from the target tissue to the NIR camera lens was also Important in order to maintain consistent fluorescence imaging and reduce variability when measuring fluorescence intensity.

### 2.3. Hyperspectral Imaging (HSI)

To assess real-time tissue oxygen saturation (StO_2_) in the colon, the TIVITA^®^ Tissue System (Diaspective Vision GmbH, Am Salzhaff, Germany) was utilized for HSI acquisition (Figure 1). HSI is a technique that captures images at multiple wavelengths, allowing for the analysis of the spectral content of the image. The HSI operates in the visible to near-infrared (VNIR) regions with approximately 400–1000 nm of the electromagnetic spectrum. In the visible range (approximately 400–750 nm), light can penetrate the surface of the colon to a depth of 1 mm. NIR spectroscopy uses NIR wavelength regions (approximately 750–1000 nm) to measure the NIR perfusion index as the status of microcirculation in tissues. NIR light can penetrate tissues to a depth of 3–4 mm [6]. A hyperspectral camera captures images at multiple wavelengths, and the spectral unmixing process separates the spectral components of oxyhemoglobin and deoxyhemoglobin. By analyzing the changes in light intensity, the device can calculate the relative amount of oxygenated and deoxygenated hemoglobin, providing an estimate tissue perfusion status. The analysis of StO_2_ and NIR perfusion index was performed using the Automatic Data Extraction & Visualization Program 2.0 (designed by the Microprocessor Application Laboratory, Pusan National University, Busan, Republic of Korea). These levels were automatically analyzed and reconstructed as a color map on the monitor screen within 15 s (Figure 2). The system includes small red and green laser sources around the main camera lens, and the operator can adjust the camera location to overlap these two colored points, ensuring a constant distance between the camera lens and the colon around 45 cm.

### 2.4. Colonic Perfusion Assessment Protocol

During the laparoscopic colorectal surgery, the inferior mesenteric artery (IMA) was ligated at a high level (high ligation) or a low level with left colic artery preservation (low ligation), depending on the cancer stage and tumor location. Then, the left-sided colon was mobilized, and the proximal colon transection was prepared by mesenteric division. After transection of the distal colorectal segment with sufficient surgical resection margin, the proximal segment of the colon was extracted from the abdominal cavity through the transumbilical mini-laparotomy site. The study protocol included the use of both ICG angiography and HSI to assess colonic perfusion status. If the colonic StO_2_ of HSI was more than 60% and the T_1/2MAX_ of ICG angiography was less than 10 s, it was considered to indicate good perfusion. Based on these criteria, colorectal anastomosis was performed on the planned transection line. If the colonic StO_2_ was less than 50% and the T_1/2MAX_ was delayed more than 25 s, it was determined as being poor perfusion, and the transection line was moved to the proximal colon and repeated perfusion evaluations were performed. If the colonic StO_2_ of the changed transection line was maintained at 60% or higher and the T_1/2MAX_ was within 25 s during the second HSI test and ICG angiography, we determined a good or intermediate perfusion status as being acceptable, and colorectal anastomosis was performed (Figure 3). The cutoff for safe StO_2_ to maintain the viability of the colon has been reported to be between 60% and 80% [8,12]. Therefore, the cutoff for safe colonic StO_2_ was set at 60% in this study.

### 2.5. Comparison of StO_2_ Levels and Quantitative Perfusion Parameters 

The StO_2_ levels were compared with the quantitative perfusion parameters derived from ICG angiography at corresponding points to define the safe range of ICG parameters reflecting good tissue oxygenation. Five points were manually selected along the center of the proximal colon segment on the StO_2_ color map of the HSI image. These points, labeled 1–5, were placed from the proximal colon towards the distal end, with point 5 always positioned on top of the transection line. Colonic areas covered with mesenteric fat and colonic appendage were avoided to ensure that the points were placed entirely over areas of the colon wall to obtain more accurate StO_2_ measurements. For the quantitative perfusion parameters of ICG angiography, the same points on the colon segment were analyzed as those selected in the StO_2_ color map of the HSI image. The ICG analysis program automatically calculated the quantitative perfusion parameters at each of these points (Figure 2).

### 2.6. Anastomotic Complications

Postoperative complications were evaluated using the Clavien-Dindo classification system. Anastomotic complications were defined as pelvic abscess, anastomotic leakage, anastomotic dehiscence, colon ischemia, and necrosis. These complications were confirmed through CT scan and sigmoidoscopy.

### 2.7. Statistical Analysis

Pearson’s Chi-square or Fisher’s exact test were used to evaluate the comparison between clinical factors and perfusion status causing transection line changes and perfusion parameters are presented mean and standard deviation. To determine the safe cutoff values for the perfusion parameters of ICG angiography, a criterion of good perfusion was set based on matching colon StO_2_ of 60% or higher. Regression analysis was used to assess the cutoff levels of the quantitative perfusion parameters reflecting tissue StO_2_ from 60% to 80%. We evaluated the validity of sensitivity, specificity, positive predictive value (PPV) and negative predictive value (NPV) using cutoff values for the perfusion parameters of ICG angiography and StO_2_ values greater than 60%. Additionally, the area under the curve (AUC) values and confidence intervals were determined by means of receiver operating characteristic (ROC) curve analysis. Network analyses were performed and heatmaps were produced based on robust Spearman correlation (rho) measures for all collected parameters. Positive (rho < 0.2) and negative (rho < −0.2) associations are represented by blue and red lines (edges), respectively. SPSS 27.0 (Statistical Package for Social Science Version 27.0, IBM SPSS, Armonk, NY, USA) and R software (version 4.3.0, R Foundation for Statistical Computing, Vienna, Austria) was used for statistical analysis and graphic, and the significance level was two-tailed, with *p* < 0.05.

## 3. Results

Laparoscopic low anterior resection (LAR, 46 cases) and anterior resection (22 cases) were performed and diverting ileostomy (24 cases) was performed in 52.2% of the LAR patients. Intraoperative perfusion assessment identified a poor perfusion segment in six patients (8.8%), and the transection line was moved to the proximal colon (Table 1).

In the group of patients (*n* = 6) for whom the transection line was changed, one patient (16.7%) experienced an anastomotic leak. Primary anastomosis was performed in the remaining 62 patients, who initially had good perfusion status of the colon. Anastomosis complications occurred in four patients (6.5%), who were initially considered to have good perfusion. Therefore, five patients (7.4%) had major anastomotic complications requiring postoperative intensive treatment. Anastomotic complications included transmural ischemia (*n* = 1), anastomotic leak (*n* = 4), and delayed pelvic abscess (*n* = 1). Reoperation was required in two cases (3.5%), and one of them happened after the liver metastasis resection following colon surgery, iatrogenic marginal artery damage and transmural ischemia of the left colon occurred due to excessive traction of the mesentery of the transverse colon, requiring subtotal resection (*n* = 1) for colonic necrosis (Figure 3).

In the regression analysis, T_1/2MAX_, T_MAX_, slope, and NIR perfusion index were correlated with tissue oxygen saturation (Figure 4). Using this regression model, the cutoff values of quantitative perfusion parameters were calculated as T_1/2MAX_ ≤ 10 s, T_MAX_ ≤ 30 s, slope ≥ 5, and NIR perfusion index ≥ 50, which best reflected colon StO_2_ higher than 60% (Table 2). However, the distributions of the perfusion TR and F_MAX_ were scattered and unpredictable to derive equation in the regression model. This non-normal distribution pattern makes it difficult to determine the suitability of the cutoff level for colon StO_2_ (≥60%) using the regression model. Therefore, the cutoff values of perfusion TR and F_MAX_ were estimated considering a 95% CI range (Table 3).

The cutoff values of T_1/2MAX_, T_MAX_, perfusion TR, slope, and NIR perfusion index were significantly related with StO_2_ (≥60%) (Table 4). Additionally, diagnostic values were analyzed to predict colon StO_2_ of 60% or more, and ICG perfusion parameters as T_1/2MAX_ and T_MAX_ showed high sensitivity values of 97% or more, indicating their ability to correctly identify cases with acceptable StO_2_. However, the perfusion TR and F_MAX_ value were limited specificity and NPV. The slope had the highest specificity and PPV to predict the acceptable tissue oxygenation (Table 5). On the ROC curve analysis, T_1/2MAX_, T_MAX_, and slope demonstrated higher area under the curve values, indicating relatively reliable predictive performance (Table 6).

Using the step-by-step analysis algorithm for quantitative ICG perfusion parameters, the prediction of tissue oxygenation was correct at 99% when both T_1/2MAX_ and T_MAX_ were within the safe range (Figure 5). When both T_1/2MAX_ and T_MAX_ were over the cutoff levels, the tissue oxygenation was significantly worse at 45.5% (*p* < 0.001). When only one of T_1/2MAX_ and T_MAX_ was within the safe range, the detecting rate of poor tissue oxygenation was 36.4% and the slope parameter was capable of accurately discriminating poor tissue oxygenation cases in these situations.

Bivariate analysis represents the correlation coefficient between the quantitative perfusion parameters of the perfusion assessment points (*n* = 340) and the corresponding clinical factors. Network analysis was performed using the correlation coefficients of the statistically correlated factors. The initial tissue oxygenation (StO_2_ < 60%) requiring transection line change demonstrated a complex correlation with several perfusion parameters, male, and old age as the critical node-forming major island. Network analysis also found complex correlations between anastomotic complications and several factors, including atherosclerotic risk score, NIR perfusion index, high ligation of IMA, old age, and preoperative chemoradiation therapy (Figure 6).

## 4. Discussion

The quantitative analysis for ICG angiography was based on the fluorescence intensity and perfusion time required for ICG to reach the target tissue and increase to the maximal fluorescence intensity. However, the criteria of the quantitative perfusion parameter for determining the adequacy of tissue oxygenation have not yet been clarified, and the normal range of ICG perfusion parameters obtained from quantitative analysis is not fully defined. This is because ICG perfusion parameters could be affected by various anatomical and physical conditions of patients, as well as other technical conditions such as the injection amount of ICG, fluorescence equipment, and other external conditions [5]. Tissue oxygenation levels are essential and reliable indicators of perfusion status for determining tissue viability [12]. Recently, the real-time StO_2_ of the colon could be measured using the HSI technique in the operating room [13]. Therefore, this study was conducted to establish a safe range of ICG perfusion parameters reflecting the adequate oxygenation level of the colon by analyzing the quantitative perfusion parameter at the point where the StO_2_ of the colon tissue was 60% or higher.

In this study, T_1/2MAX_ and T_MAX_ adequately reflected the tissue oxygenation levels of the colon. The safe ranges of T_1/2MAX_ and T_MAX_ for adequate StO_2_ (≥60%) were analyzed as being within 10 s and 30 s, respectively. In previous studies, the cutoff of T_1/2MAX_ for predicting the risk of anastomotic complications has been suggested to be 18 s. In this study, the cutoff for predicting safe tissue StO_2_ (≥60%) was 10 s. However, T_1/2MAX_ existed between 10 and 25 s at some perfusion sites where tissue StO_2_ was maintained above 60%. Although this was classified as an intermediate perfusion status, postoperative anastomotic complications did not occur. 

Indeed, it is essential to consider various factors that can influence T_1/2MAX_ and T_MAX_ values, even in the colon, where adequate StO_2_ is maintained. In this study, prolongation of T_1/2MAX_ and T_MAX_ occurred even when tissue oxygenation was adequate, which occurred when the ICG dose used in the patient was insufficient, or when the patient’s systemic systolic blood pressure temporarily dropped below 70 mmHg at the time of perfusion assessment. During general anesthesia, fluctuations in cardiovascular conditions such as hypotension and vasoconstriction can compromise mesenteric blood flow and subsequently affect T_1/2MAX_ values. Certain patient conditions, such as hypovolemia, anemia, or pre-existing vascular diseases, can also contribute to altered perfusion parameters [2]. In addition, the patient’s excessive posture change, systemic blood pressure, injected ICG volume, distance of camera to tissue, fluorescence camera system, and type of light source as Xenon, light-emitting diode (LED), or laser should be considered to interpret the ICG perfusion parameter. It is important for surgeons to interpret T_1/2MAX_ values cautiously in conjunction with the patient’s clinical and overall hemodynamic status. While a cutoff of 10 s may indicate safe tissue oxygenation in this study, the potential confounding factors should be considered when assessing perfusion parameters. Therefore, even if T_1/2MAX_ and T_MAX_ are prolonged, it should be considered that other perfusion parameter and clinical factors such as the bowel color, peristalsis, and arterial pulsation for determining the perfusion status by surgeon.

A perfusion TR of 0.6, the cutoff in a previous study, was associated with anastomotic complications. However, the range of perfusion TR reflecting tissue oxygenation in this study required more careful attention. When the blood flow was insufficient, T_MAX_ and T_1/2MAX_ were simultaneously delayed, and the perfusion TR showed the same result as normal blood flow. Therefore, the interpretation of T_1/2MAX_ and T_MAX_ should be prioritized over perfusion TR. In a previous study, T_1/2MAX_ showed the highest sensitivity, while perfusion TR showed the highest specificity to predict anastomosis complications [1]. Therefore, it is necessary to review the various perfusion parameters of ICG angiography using step by step algorithm of quantitative analysis to predict tissue oxygenation level and the risk of anastomosis complications.

F_MAX_ is an indicator of fluorescence intensity and is the parameter most affected by external environmental factors [5]. In this study, despite the application of the standardized ICG angiography protocol, a scattered distribution was observed, and there was no correlation with tissue StO_2_ in the regression model. Nevertheless, the increasing slope, which simultaneously reflects perfusion time and fluorescence intensity, was significantly correlated with tissue StO_2_. When the ICG angiography image was obtained using an open NIR camera system, the slope was found to be the most significant perfusion factor reflecting the colonic blood flow [14]. This is presumed to be due to the difference in fluorescence light intensity between the laparoscopic and open NIR cameras. The use of several powerful LED lights in the open NIR camera can minimize the variation errors caused by external confounding factors. Intensified LED lights provide a stable and consistent source of illumination, which can result in a reliable fluorescence intensity [15]. Therefore, it is possible that the values of F_MAX_ and slope may be underestimated in this study due to the use of a laparoscopic NIR camera, which typically uses a weak fluorescence light source within a low-caliber scope. This limitation can result in a lower recorded F_MAX_ and potentially affect the accuracy of the slope. If future advancements in technology lead to improvements in fluorescence light intensity and the light source used in laparoscopic NIR cameras, it is likely that fluorescence intensity measurements will become more stable and reliable. This would minimize the variations caused by external confounding factors and enhance the accuracy and reproducibility of the ICG perfusion parameters.

The quantitative perfusion parameters of ICG angiography, such as perfusion time or fluorescence intensity, do not directly reflect tissue StO_2_. While both ICG angiography and HSI provide valuable information about tissue perfusion, they measure different aspects of tissue perfusion and are not considered interchangeable [16]. ICG angiography captures the immediate transit of the fluorescence dye through the vasculature, so the ICG perfusion parameter may not perfectly match with the tissue oxygenation status. Additionally, tissue oxygenation levels can be influenced by factors such as oxygen consumption, local metabolism, and tissue perfusion dynamics, which may not be directly reflected in ICG perfusion parameters [17,18,19]. Therefore, the measurement of tissue StO_2_ will be beneficial for assessing whether adequate blood flow reaches tissue microcirculation levels [20]. However, as the commercial prevalence and clinical application of HSI are quite limited to date, the StO_2_ of the colon is not routinely measured as part of standard diagnostic evaluations in clinical practice. Recently, laparoscopic NIR camera systems have rapidly become popular in the surgical field. Therefore, the development and validation of more accurate and reliable perfusion parameter of ICG angiography is an urgent task at present [21,22,23]. In particular, the verification of ICG perfusion parameters that reflect the appropriate oxygenation to maintain colon viability can help surgeons make critical decisions for colonic perfusion status and surgical strategy in the operating room [24]. Therefore, the strength of this study is that it compared perfusion parameters from ICG angiography to tissue oxygenation from HSI simultaneously. In this aspect, this study could provide valuable safe values of ICG perfusion parameters to predict adequate oxygen supply for colon tissue viability.

A wide range of StO_2_ values in the colon were reported, ranging from approximately 20% to 80% [12]. However, it is difficult to establish a definitive normal range for StO_2_ in the colon because of the variability in measurement methods and lack of standardized criteria for interpreting the results. Therefore, the StO_2_ level required for the colon viability is not precisely known. Generally, StO_2_ in the colon is lower than that in other parts of the body because of the presence of colonic bacteria that consume oxygen during fermentation. The range of StO_2_ in a healthy colon can vary depending on individual factors; however, the normal StO_2_ level in the colon is typically considered between 60% and 80%. The minimum StO_2_ level at which colonic ischemia can occur has not been well established. Some studies have suggested that the colon typically requires an StO_2_ level of at least 50% for adequate tissue perfusion and function [25]. Another study reported that a reduction in oxygen delivery to the colon to 50% of baseline for a persistent period could result in colonic ischemia [26]. In this study, the StO_2_ cutoff was set at 60% to define the safe cutoff values of the ICG perfusion parameters. Interestingly, when colon StO_2_ was below 50%, significant delays in T_1/2MAX_ and T_MAX_ were observed. Therefore, delays of T_1/2MAX_ and T_MAX_ outside the safe range should be considered a potential indication of compromised tissue oxygenation below 50%.

### 4.1. Study Limitations

This study has several limitations. First, this study was conducted in a single institution and with a small patient group. This observational study can provide valuable preliminary data, but further research with larger and more diverse patient cohorts is needed to validate and strengthen the findings. Second, discrepancies between the ICG perfusion parameter and tissue StO_2_ were observed at certain points of perfusion assessments. These discrepancies can arise from various external factors and limitations of each measurement technique. During the acquisition of ICG angiography, patient respiration and minute shaking on the part of the examiner could create respiration-induced motion artifacts that could affect the alignment and registration of images acquired at different points, leading to discrepancies when comparing specific points of interest between ICG angiography and the StO_2_ level of HSI [27]. This limitation can make it difficult to achieve a precise and accurate comparison between the same points in the ICG and HSI images. Third, the fluorescence intensity can be influenced by various external confounding factors, especially the intensity and type of light source of the NIR camera system. In this study, a laparoscopic NIR camera system equipped with a weak intensity of fluorescence and laser light source. Consequently, the cutoff values could not directly be applied to other NIR camera systems that utilize different intensity and type of fluorescence light source. Fourth, tissue oxygenation assessment did not demonstrate an effect in reducing anastomotic complications in this study. Anastomosis complications still existed even after optimizing tissue oxygenation using ICG angiography and HSI. The effectiveness of ICG angiography in reducing anastomotic complications remains controversial [28]. The causes of anastomosis complications are combined by various clinical factors and hypoperfusion or poor tissue oxygenation must be one of critical issues [29]. This study identified anastomotic complications as complex network core nodes, indicating that multiple clinical factors contribute to these complications. Interestingly, the need to change the transection line due to poor initial StO_2_ was not found to be correlated with anastomosis complications. This suggests that adjusting the transection line based on initial perfusion status may help reduce the risk of anastomotic complications caused by inadequate blood supply to the colon. Studies focusing on optimizing perfusion have reported a significantly lower incidence of anastomotic complications [30]. This highlights the potential importance of evaluating blood flow during colorectal surgery to decrease complications [31]. Therefore, it is expected that perfusion assessment should become an essential surgical procedure for ensuring patient safety in colorectal surgery [32].

### 4.2. Conclusions

The safe values for quantitative perfusion parameters were derived from ICG angiography as T_1/2MAX_ ≤ 10 s and T_MAX_ ≤ 30 s, which were associated with colon tissue oxygenation levels higher than 60% in the laparoscopic colorectal surgery.

## Figures and Tables

**Figure 1 biomedicines-11-02029-f001:**
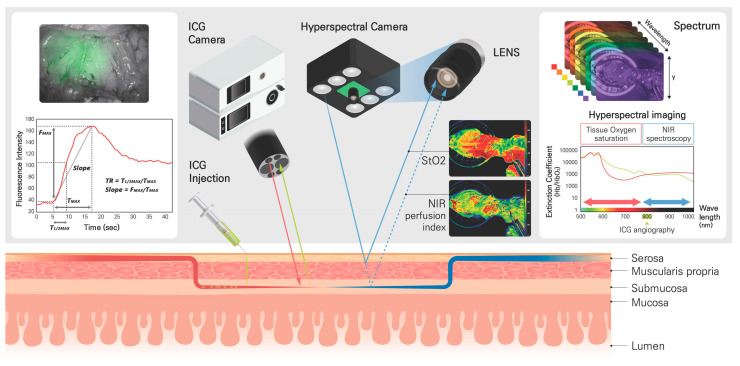
Schematic diagram of colon perfusion assessment using indocyanine green (ICG) angiography and hyperspectral imaging (HSI). Intravenous ICG binds to intravascular globulin or albumin and remains in the vascular circulation. A near-infrared (NIR) ray with a wavelength of 806 nm (red line) is emitted from the laparoscopic camera, causing ICG in the blood vessel to emit a wavelength of 830 nm (green line), and the fluorescence image is displayed on the monitor. The quantitative analysis for ICG angiography is based on the fluorescence intensity and perfusion time required for ICG to reach the target tissue and increase to the maximal fluorescence intensity as a time–fluorescence intensity graph. The perfusion time factors are calculated as T_MAX_ (time from first fluorescence increase to maximum), T_1/2MAX_ (time to reach half of the maximum), and perfusion TR (time ratio, T_1/2MAX_/T_MAX_). The fluorescence intensity factors are F_MAX_ (maximal intensity) and slope (calculated as ΔF/ΔT = F_MAX_/T_MAX_). HSI is a technique that captures images at multiple wavelengths from Xenon lamp (blue line) in the visible to NIR regions with approximately 400–1000 nm of the electromagnetic spectrum. The StO_2_ (blue line) and NIR perfusion index (blue dotted line) are calculated by analyzing the changes in light intensity. The device can calculate the relative amount of oxygenated and deoxygenated hemoglobin, providing an estimate of perfusion status at colon depths of 1 mm and 3–4 mm.

**Figure 2 biomedicines-11-02029-f002:**
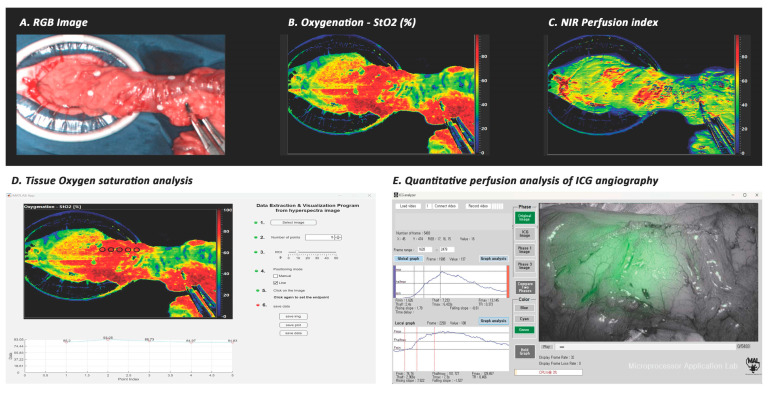
Quantitative analysis of colonic perfusion assessment using ICG angiography and HSI imaging: (**A**) RGB image of HSI imaging; (**B**) color map of StO_2_; (**C**) color map of NIR perfusion index; (**D**) tissue StO_2_ analysis on the five points (black circles) along the proximal colon segment using the Automatic Data Extraction & Visualization Program 2.0 (designed by the Microprocessor Application Laboratory, Pusan National University, Busan, Republic of Korea); (**E**) quantitative perfusion analysis of ICG angiography using the ICG Analyzer Program 8.0 (designed by the Microprocessor Application Laboratory, Pusan National University, Busan, Republic of Korea).

**Figure 3 biomedicines-11-02029-f003:**
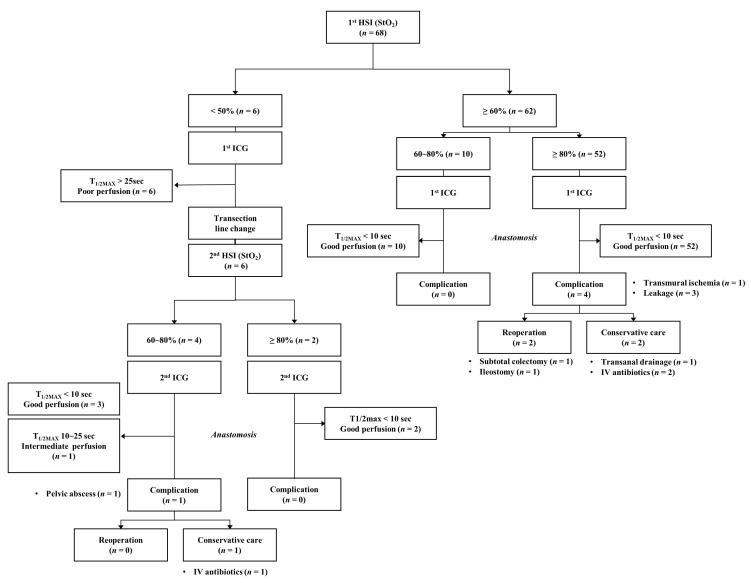
Flowchart of colonic perfusion assessment protocol using ICG angiography and HSI imaging. If the colonic StO_2_ is greater than 60% and the T_1/2MAX_ of ICG angiography is less than 10 s, colorectal anastomosis is performed on the planned transection line with favorable perfusion. If the colonic StO_2_ is less than 50% and the T_1/2MAX_ is delayed more than 25 s, it is determined to be poor perfusion, and the transection line is changed to the proximal colon, and second perfusion evaluations are performed.

**Figure 4 biomedicines-11-02029-f004:**
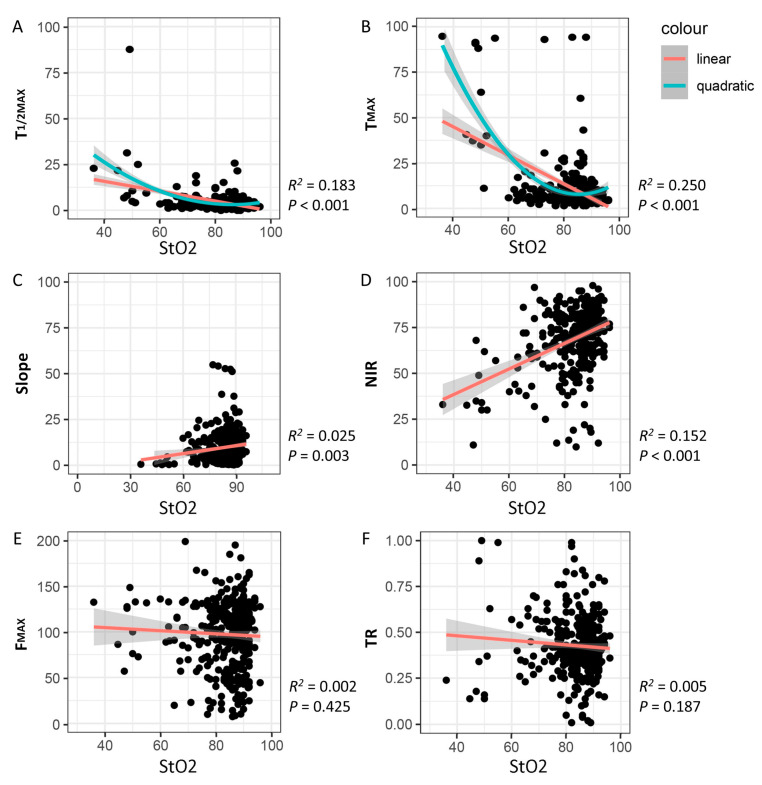
Scatter plot of quantitative perfusion parameters and colonic StO_2_ in regression analysis. (**A**) T_1/2MAX_, (**B**) T_MAX_, (**C**) slope, and (**D**) NIR perfusion index exhibit significant correlations with colonic StO_2_. However, the distributions of (**E**) F_MAX_ and (**F**) perfusion TR are currently distracting and unpredictable, making it challenging to derive an equation in the regression model. The orange line represents the regression line of the linear model, while the green line represents the regression line of the quadratic model.

**Figure 5 biomedicines-11-02029-f005:**
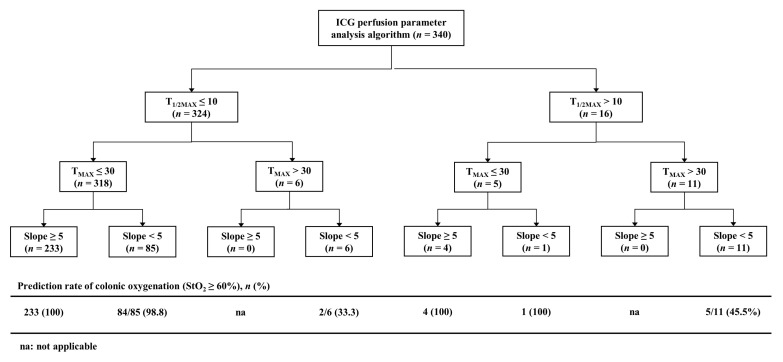
Step-by-step analysis algorithm for quantitative ICG perfusion parameters (T_1/2MAX_, T_MAX_, and slope) for the prediction of colonic oxygenation (StO_2_ ≥ 60%).

**Figure 6 biomedicines-11-02029-f006:**
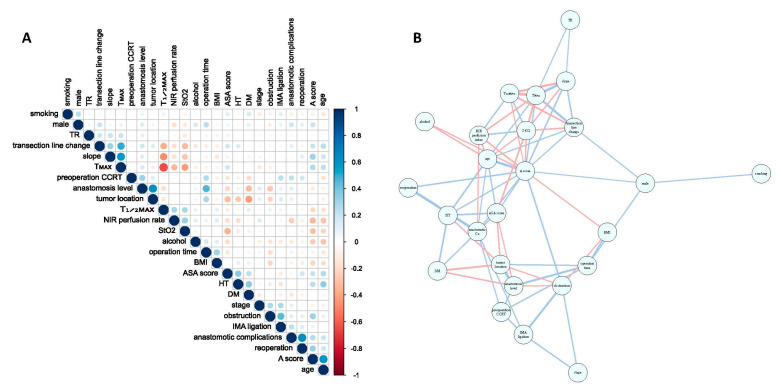
Correlation plot and network analysis. (**A**) Bivariate analysis represents the Spearman correlation coefficient between the quantitative perfusion parameters of the perfusion assessment points (*n* = 340) and the corresponding clinical factors. (**B**) Network analysis of Spearman’s coefficients currently shows a complex correlation link between transection line change and various clinical factors. The blue line reflects significant correlation with a coefficient > 0.2, while the red line reflects reverse correlation with a coefficient < −0.2. A score, atherosclerotic risk score; Cx, complication.

**Table 1 biomedicines-11-02029-t001:** Patients’ characteristics.

Clinical Factor	Initial StO_2_, *n* (%)		*p*-Value ^†^
	StO_2_ (≥60%), *n* = 62	StO_2_ (<60%), *n* = 6	
Age (≥70 yr)	25 (40.3)	5 (83.3)	0.080
Male:Female	41:21	6:0	0.186
BMI (≥25 kg/m^2^)	22 (35.5)	1 (16.7)	0.856
ASA score (≥3)	7 (11.3)	2 (33.3)	0.177
Hypertension	26 (41.9)	4 (66.7)	0.394
Diabetes	13 (21.0)	1 (16.7)	1.000
Smoking	7 (11.3)	1 (16.7)	0.543
Alcohol drinking	13 (100)	0 (0)	0.587
Total cholesterol (≥220 mg/dL)	11 (17.7)	0 (0)	0.579
Atherosclerotic risk score (≥20)	12 (19.4)	3 (50.0)	0.116
Stage (≥III)	26 (41.9)	2 (33.3)	1.000
Cancer location			
Sigmoid	25 (96.2)	1 (3.8)	0.395
Rectum	37 (88.1)	5 (11.9)	
PCRT	8 (12.9)	1 (16.7)	1.000
Cancer obstruction	14 (22.6)	1 (16.7)	1.000
Operation (LAR)	41 (66.1)	5 (83.3)	0.656
Anastomotic level (<5 cm)	22 (35.5)	4 (66.7)	0.193
Diverting ileostomy	20 (32.3)	4 (66.7)	0.175
IMA ligation (high)	36 (58.1)	4 (66.7)	1.000
Anastomotic complication	4 (6.5)	1 (16.7)	0.379
Reoperation	2 (3.2)	0	1.000

BMI, body mass index; ASA, American Society of Anesthesiologists; PCRT, preoperative chemoradiation therapy; LAR, low anterior resection; IMA, inferior mesenteric artery; ^†^, Fisher’s exact test.

**Table 2 biomedicines-11-02029-t002:** Regression analysis for cutoff level of quantitative perfusion parameters.

	Regression Analysis			Cutoff Value, Mean (95% CI)				
	Model (n)	Equation	*R^2^*	StO_2_ 60%	StO_2_ 65%	StO_2_ 70%	StO_2_ 75%	StO_2_ 80%
T_1/2MAX_ (s)	linear (*n* = 340)	T_1/2MAX_ = 26.351 − 0.267 × StO_2_	0.183	10.31	8.98	7.64	6.30	4.97
				(8.79, 11.83)	(7.73, 10.22)	(96.66, 8.63)	(5.55, 7.06)	(4.37, 5.57)
	quadratic (*n* = 340)	T_1/2MAX_ = 82.801 − 1.851 × StO_2_ + 0.011(StO_2_)^2^	0.260	10.40	7.86	5.85	4.39	3.46
				(8.95, 11.85)	(6.61, 9.10)	(4.74, 6.97)	(3.42, 5.35)	(2.70, 4.22)
T_MAX_ (s)	linear (*n* = 340)	T_MAX_ = 76.287 − 0.785 × StO_2_	0.250	29.18	25.25	21.33	17.40	13.48
				(25.52, 32.84)	(22.26, 28.25)	(18.96, 23.70)	(15.58, 19.22)	(12.03, 14.92)
	quadratic (*n* = 340)	T _MAX_ = 252.715 − 5.736 × StO_2_ + 0.034(StO_2_)^2^	0.369	29.45	21.76	15.74	11.41	8.76
				(26.09, 32.81)	(18.87, 24.64)	(13.17, 18.32)	(9.18, 13.65)	(6.99, 10.52)
Slope (s/AU)	linear (*n* = 340)	Slope = −2.236 + 0.144 × StO_2_	0.025	6.38	7.10	7.82	8.54	9.26
				(3.98, 8.79)	(5.13, 9.07)	(6.26, 9.38)	(7.34, 9.73)	(8.31, 10.20)
	linear (*n* = 335)	Slope = −3.087 + 0.146 × StO_2_	0.043	5.68	6.41	7.14	7.87	8.60
				(3.81, 7.55)	(4.88, 7.94)	(5.92, 8.35)	(6.93, 8.80)	(7.86, 9.34)
NIR perfusion index	linear (*n* = 340)	NIR = 10.515 + 0.699 × StO_2_	0.152	52.48	55.98	59.48	62.98	66.47
				(48.05, 56.92)	(52.35, 59.62)	(56.60, 62.35)	(60.77, 65.19)	(64.72, 68.23)
	linear (*n* = 332)	NIR = 13.125 + 0.683 × StO_2_	0.185	54.12	57.54	60.96	64.37	67.79
				(50.22, 58.03)	(54.35, 60.73)	(58.43, 63.48)	(62.44, 66.31)	(66.27, 69.31)

AU, arbitrary unit.

**Table 3 biomedicines-11-02029-t003:** Cutoff level of quantitative perfusion parameters for tissue oxygenation (StO_2_ ≥ 60%).

Perfusion Parameter	Mean ± SD	Range	95% CI of Mean	Cutoff Level (StO_2_ ≥ 60%)
T_1/2MAX_ (s)	4.07 ± 5.85	0.03–87.67	3.45–4.70	≤10 s ^a^
T_MAX_ (s)	10.85 ± 14.68	1.57–94.77	9.28–12.41	≤30 s ^a^
Perfusion TR	0.43 ± 0.16	0.01–1.00	0.41–0.44	≤0.8 ^b^
F_MAX_ (AU)	97.21 ± 37.00	7.62–199.01	93.26–101.16	≥25 ^b^
Slope (AU/s)	9.74 ± 8.46	0.33–54.95	8.83–10.64	≥5 ^a^
NIR perfusion index	68.82 ± 16.76	10.0–98.0	67.03–70.60	≥50 ^a^

SD, standard deviation; CI, confidence interval; ^a^, cutoff value derived from regression model; ^b^, cutoff value estimated considering a 95% CI range.

**Table 4 biomedicines-11-02029-t004:** Comparisons of quantitative perfusion parameters with tissue oxygenation (StO_2_ ≥ 60%).

Perfusion Parameter	Cut Off Level	StO_2_, *n* (%)		*p*-Value
		<60%	≥60%	
T_1/2MAX_ (s)	≤10	5 (1.5)	319 (98.5)	<0.001
	>10	6 (37.5)	10 (62.5)	
T_MAX_ (s)	≤30	1 (0.3)	322 (99.7)	<0.001
	>30	10 (58.8)	7 (41.2)	
Perfusion TR	≤0.8	8 (2.4)	322 (97.6)	<0.001
	>0.8	3 (30.0)	7 (70.0)	
F_MAX_ (AU)	≤25	0 (0)	15 (100)	0.469
	>25	11 (3.4)	314 (96.6)	
Slope (s/AU)	<5	11 (10.4)	95 (89.6)	<0.001
	≥5	0 (0)	234 (100)	
NIR perfusion index	<50	8 (18.6)	35 (81.4)	<0.001
	≥50	3 (1.0)	294 (99.0)	
Total		11 (3.2)	329 (96.8)	

**Table 5 biomedicines-11-02029-t005:** Diagnostic values of quantitative ICG perfusion parameters for predicting tissue oxygenation (StO_2_ ≥ 60%).

Perfusion Parameter	Cutoff Level	Sensitivity	Specificity	PPV	NPV	Accuracy
T_1/2MAX_ (s)	≤10	97.0	54.5	98.5	37.5	95.6
		(94.5, 98.5)	(23.4, 83.3)	(96.4, 99.5)	(15.2, 64.6)	(92.8. 97.5)
T_MAX_ (s)	≤30	97.9	90.9	99.7	58.8	97.6
		(95.7, 99.1)	(58.7, 99.8)	(98.3, 100)	(32.9, 81.6)	(95.4, 98.9)
Perfusion TR	≤0.8	97.9	27.3	97.6	30.0	95.6
		(95.7, 99.1)	(6.0, 61.0)	(95.3, 98.9)	(6.7, 65.2)	(92.8, 97.5)
F_MAX_ (AU)	>25	95.4	0	96.6	0	92.4
		(96.2, 99.4)	(0, 28.5)	(94.6, 98.3)	(0, 21.8)	(89.0, 94.9)
Slope (AU/s)	≥5	71.1	100	100	10.4	72.1
		(65.9, 76.0)	(71.5, 100)	(98.4, 100)	(5.3, 17.8)	(67.0, 76.8)

PPV, positive predictive value; NPV, negative predictive value; Diagnostic values expressed as mean (95% CI).

**Table 6 biomedicines-11-02029-t006:** Receiver operating characteristic (ROC) curve analysis.

Perfusion Parameter	AUC	95% CI	*p*-Value
T_1/2MAX_ (s)	0.944	0.894–0.994	<0.001
T_MAX_ (s)	0.976	0.949–1.000	<0.001
Perfusion TR	0.428	0.176–0.679	0.415
F_MAX_ (AU)	0.404	0.234–0.579	0.277
Slope (AU/s)	0.948	0.905–0.991	<0.001

AUC, area under the curve.

## Data Availability

The data presented in this study are available on request from the corresponding author. The data are not publicly available due to ethical and privacy reasons.

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
