# Peer review of "The Safe Values of Quantitative Perfusion Parameters of ICG Angiography Based on Tissue Oxygenation of Hyperspectral Imaging for Laparoscopic Colorectal Surgery: A Prospective Observational Study"

_biomedicines, 2023, doi:10.3390/biomedicines11072029_

Round 1
Reviewer 1 Report
Thank you for involving me in this review.
A very innovative and current topic that has been evolving in recent years with the final aim of lowering the percentage of anastomotic leakage even more.
Fluent in English and easily understood.
Statistical analysis well conducted and quite understandable.
Good quality images and graphics.
Small changes I would recommend:
- In the abstract, the "AIM" section must be replaced with "Background" and before starting immediately with the purpose of the work, write at least one introductory sentence on the subject in general;
- Also in the abstract the acronym HIS has not been specified.
Reviewer 2 Report
Son et al. the optimal method of assessing tissue perfusion before colon resection in order to prevent surgical complications. The manuscript was scientifically prepared with a series of well-designed experiments. This topic is relevant to their field of expertise. Real-time tissue oxygen saturation (StO2) in the colon is measured by hyperspectral imaging (HSI). are analyzed. During HSI, images are captured at multiple wavelengths that operate in the visible near-infrared (NIR) range. NIR light can penetrate the tissues to a depth of 3-4 mm, and based on the NIR perfusion index, the state of tissue microcirculation can be deduced. The study protocol included ICG angiography and HSI to assess colonic perfusion status.
- If HSI colonic StO2 is greater than 60% and ICG angiography T1/2MAX is less than 10 seconds, this indicates good perfusion. Based on these criteria, a colorectal anastomosis was performed on the planned transection line.
- If the colonic StO2 was less than 50% and T1/2MAX was delayed more than 25 seconds, poor perfusion was assumed and the transection line was placed in the more proximal colon and repeated perfusion studies were performed until the ideal location was found. with good tissue perfusion to avoid postoperative complications (e.g. lack of sutures, abscess).
The discussion is well organized and the references are adequate.
The conclusion is a bit clear, it could be more detailed.
My question is: how can we use it in everyday practice?
Reviewer 3 Report
This is a very well-designed and elegantly presented study aimed at determining the safe values for quantitative perfusion parameters obtained through indocyanine green angiography by evaluating tissue oxygenation levels using hyperspectral imaging during laparoscopic colorectal surgery. The study found that the safe quantitative perfusion parameters derived from ICG angiography, specifically T1/2MAX (≤ 10 sec) and TMAX (≤ 30 sec), were associated with colon tissue oxygenation levels higher than 60% in laparoscopic colorectal surgery. The study also found that indocyanine green perfusion parameters such as T1/2MAX, TMAX, and perfusion TR exhibited high sensitivity values exceeding 97%, indicating their capability to identify cases with acceptable StO2 levels accurately. This is a very interesting study from both basic science and clinical point of view; I hope the authors will continue this study and present a paper correlating the particular kind of lesion with parameters obtained using this method to obtain a tool for diagnostic and prognostic purposes.
